# Integrated Transcriptome and Proteome Analysis Revealed the Regulatory Mechanism of Hypocotyl Elongation in Pakchoi

**DOI:** 10.3390/ijms241813808

**Published:** 2023-09-07

**Authors:** Xiaofeng Li, Dandan Xi, Lu Gao, Hongfang Zhu, Xiuke Yang, Xiaoming Song, Changwei Zhang, Liming Miao, Dingyu Zhang, Zhaohui Zhang, Xilin Hou, Yuying Zhu, Min Wei

**Affiliations:** 1State Key Laboratory of Crop Biology, College of Horticulture Science and Engineering, Shandong Agricultural University, Tai’an 271018, China; lixiaofeng@saas.sh.cn; 2Shanghai Key Laboratory of Protected Horticultural Technology, Horticultural Research Institute, Shanghai Academy of Agricultural Sciences, Shanghai 201403, China; ddxi@saas.sh.cn (D.X.); gaolu@saas.sh.cn (L.G.); zhuhongfang@saas.sh.cn (H.Z.); yxk@163.com (X.Y.); 11616010@zju.edu.cn (L.M.); zhangdy1225@126.com (D.Z.); szyzzh@163.com (Z.Z.); 3College of Horticulture, Nanjing Agricultural University, Nanjing 210095, China; changweizh@njau.edu.cn (C.Z.); hxl@njau.edu.cn (X.H.); 4College of Life Sciences, North China University of Science and Technology, Tangshan 063210, China; songxm@ncst.edu.cn

**Keywords:** hypocotyl, transcriptome, proteome, hormone, photosynthesis, pakchoi

## Abstract

Hypocotyl length is a critical determinant for the efficiency of mechanical harvesting in pakchoi production, but the knowledge on the molecular regulation of hypocotyl growth is very limited. Here, we report a spontaneous mutant of pakchoi, *lhy7.1*, and identified its characteristics. We found that it has an elongated hypocotyl phenotype compared to the wild type caused by the longitudinal growth of hypocotyl cells. Different light quality treatments, transcriptome, and proteomic analyses were performed to reveal the molecular mechanisms of hypocotyl elongation. The data showed that the hypocotyl length of *lhy7.1* was significantly longer than that of WT under red, blue, and white lights but there was no significant difference under dark conditions. Furthermore, we used transcriptome and label-free proteome analyses to investigate differences in gene and protein expression levels between *lhy7.1* and WT. At the transcript level, 4568 differentially expressed genes (DEGs) were identified, which were mainly enriched in “plant hormone signal transduction”, “photosynthesis”, “photosynthesis–antenna proteins”, and “carbon fixation in photosynthetic organisms” pathways. At the protein level, 1007 differentially expressed proteins (DEPs) were identified and were mainly enriched in photosynthesis-related pathways. The comprehensive transcriptome and proteome analyses revealed a regulatory network of hypocotyl elongation involving plant hormone signal transduction and photosynthesis-related pathways. The findings of this study help elucidate the regulatory mechanisms of hypocotyl elongation in *lhy7.1*.

## 1. Introduction

Pakchoi (*Brassica*. *Campestris* var. *communis* Tesn et Lee) has been cultivated in southern China for more than 1600 years [1] and is now an important vegetable worldwide for its sweet, succulent, and nutritious leaves and stalks [2,3]. It is a type of non-heading Chinese cabbage and can be further divided into white petiole and green petiole types. At present, the farmed area of pakchoi in China is around 1.3333 million hm^2^, which is important to the annual production and supply of vegetables [4]. With an increased planting area and aging of employees, harvesting shortages are prominent during pakchoi production due to inefficient manual harvesting. Therefore, it is necessary to popularize mechanical harvesting which needs longer hypocotyl length pakchoi cultivars to improve work efficiency. Consequently, it is essential to develop or select pakchoi cultivars with longer hypocotyls as the common pakchoi cultivars exhibit short hypocotyls.

Hypocotyl is the embryonic stems connecting the cotyledons and radicles and a channel for the transportation of signaling molecules such as water, inorganic salts, organic nutrients, and plant hormones, which makes it of great significance to the life extension of dicotyledonous plants [5]. Hypocotyl is a very plastic organ and is strongly influenced by endogenous genetic and environmental factors, such as light, gravity, temperature, shade, inorganic ions, organic compounds, and plant hormones [6]. After seed germination out of the soil, seedlings undergo light-induced photomorphogenesis associated with an inhibition of hypocotyl elongation, opening of apical hook, expansion of cotyledons, and accumulation of chlorophyll [6]. Plants perceive lights by a suite of photoreceptors including phytochromes (PHYA-E for red and far-red light), cryptochromes and phytotropins (CRY1 and 2 for blue/UVA light), and UVR8 (UV RESISTANCE LOCUS 8 for UV-B) to modulate photomorphogenesis [7,8,9,10,11,12]. Upon light conditions, photoreceptors interact with COP1, a key RING-finger-E3 ubiquitin ligase, to inhibit the degradation of COP1 substrates, including HY5/HYH, and further lead to hypocotyl elongation [13,14]. Moreover, the bHLH transcription factors, PHYTOCHROME-INTERACTING FACTORS (PIFs), act downstream of phyB and are subsequently degraded to mediate light signaling to repress photomorphogenesis [14]. In the presence of UV-B, UVR8 interacts with COP1 and promotes PIF4/5 degradation, resulting in HY5 and an HY5-dependent induction of genes to inhibit hypocotyl elongation [15].

Despite light signaling, previous studies showed that most plant endogenous hormones, such as auxin, gibberellin, ethylene, abscisic acid, brassinolide, jasmonic acid (JA), etc., can regulate the elongation of plant hypocotyls [16,17]. Light regulates hypocotyl elongation in part through auxin pathways. Three auxin response factors, ARF6, ARF7, and ARF8, together promote hypocotyl elongation [17]. Under red or blue light, phyB and CRY1 can interact with ARF6 and ARF8, respectively [14]. After interaction, ARF activities are repressed to negatively affect auxin signaling. Moreover, ethylene inhibits hypocotyl elongation in yellowing seedlings during darkness and promotes hypocotyl elongation by inducing cell expansion under light conditions [18]. Cytokinin inhibits hypocotyl elongation in darkness through the ethylene pathway and promotes elongation under light when ethylene is blocked [19]. Brassinolide promotes cell division and elongation, regulates the arrangement of microtubules, changes the properties of cell walls, increases cell permeability and water absorption, and promotes cell expansion to affect hypocotyl elongation [20]. In summary, hypocotyl elongation is regulated by integrated signals from light, phytohormones, and other inputs, but not a single stimulus.

Recently, with the iteration and development of high-throughput sequencing technology, transcriptome analysis has become an important tool for studying complex biological processes at the molecular level and identifying candidate genes involved in growth, development, and responses to the stresses of many plants. Since biological processes are ultimately controlled by proteins, proteome research reflects the status of organisms under specific conditions [21]. Due to the incomplete agreement between biological events described at the gene level and protein level, the integration of transcriptome and proteome analyses is a better approach and is extensively applied to elucidate physiological regulatory mechanisms from different biological perspectives [22,23]. A conjoint transcriptome and proteome analysis revealed the roles of exogenous sulfur in regulating glucosinolate synthesis in cabbage and the molecular mechanism of responding to salt stress during seed germination in hulless barley [24,25]. However, there is little research on the molecular mechanisms of hypocotyl elongation using the integration of transcriptome and proteome analyses in pakchoi.

In this study, a spontaneous long hypocotyl mutant *lhy7.1* and a short hypocotyl wild type (WT) were used as experimental materials for the analysis of hypocotyl elongation by integrating transcriptome and proteome analyses. *lhy7.1* displayed longer hypocotyls under white light conditions, appearing to have a repression of photomorphogenesis compared to WT. Microscopic observations showed that the cell length of *lhy7.1* was longer than that of WT, indicating that the longer hypocotyl was due to the elongation of cells. The combined analysis of the transcriptome and proteome revealed that the auxin- and light-signaling associated genes or proteins had altered expression levels, implying their contributions to the elongated hypocotyl of *lhy7.1*. This work contributes to furthering our understanding of the complexity of hypocotyl elongation and has important theoretical and practical significance for the further breeding of long-hypocotyl varieties of pakchoi.

## 2. Results

### 2.1. Phenotypic Characterization of lhy7.1Mutant

A spontaneous hypocotyl mutant *lhy7.1* was identified from an inbred line WT of pakchoi. We analyzed the cotyledon colors, true leaf colors, and hypocotyl lengths seven days after the sowing (DAS) of mutant *lhy7.1* and WT in the artificial climate box under the optimal ambient temperature (Figure 1). The results showed that the cotyledon colors (Figure 1A) and true leaf colors (Figure 1B) of *lhy7.1* and WT were green. The average hypocotyl length of *lhy7.1* was 55.67 mm, significantly longer than WT with an average length of 30.76 mm (Figure 1C,D).

We further examined the dynamic hypocotyl growth of *lhy7.1* and WT under white light. The hypocotyl lengths of both *lhy7.1* and WT increased gradually until eight DAS. The average hypocotyl length of *lhy7.1* was significantly longer than WT after every 24 h (Figure 1E). Similarly, the hypocotyl elongation rate in *lhy7.1* was significantly higher than that of WT at any time point, which reached its peak at 72 h after sowing. The elongation rate in *lhy7.1* was the fastest at 120 h after sowing, while the elongation rate in WT was the fastest at 96 h after sowing. Subsequently, the elongation rate in *lhy7.1* and WT gradually decreased to a stable state (Figure 1F).

To identify the underlying cytological basis for the long hypocotyl phenotype of *lhy7.1*, semi-thin sections of ten DAS hypocotyls of WT and *lhy7.1* seedlings were analyzed (Figure 1G–I). Under a microscope, the length of the hypocotyl cells of *lhy7.1* was significantly longer than that of WT (Figure 1G,H). The average cell lengths of *lhy7.1* and WT were 388.98 μm and 238.57 μm, respectively (Figure 1I). These results suggest that the longer hypocotyl in *lhy7.1* was primarily caused by longitudinal cell elongation, consistent with a previous report that hypocotyl growth mainly depends on cell longitudinal elongation rather than cell division after germination [26].

### 2.2. Different Light Quality Affects Hypocotyl Growth

We further investigated the hypocotyl growth of the mutant *lhy7.1* and WT under white, red, and blue lights, as well as under dark conditions (Figure 2). The hypocotyl length of *lhy7.1* was significantly longer than that of WT under red, blue, and white lights; there was no significant difference under dark conditions (Figure 2A,B). Under red conditions, both hypocotyl lengths of WT and *lhy7.1* were longer compared to those under white light. (Figure 2B). However, in blue light, the hypocotyl length of *lhy7.1* was similar to that in white light (Figure 2B), indicating that it was mainly red signaling that was affected in *lhy7.1*. In addition, most *lhy7.1* cotyledons partially unfolded six DAS, while WT cotyledons had already rapidly unfolded after excavation, exhibiting a phenotype of partially dark morphogenesis (Figure 2C). These results suggested that *lhy7.1* was defective in the repression of the photomorphogenesis during development in red and white light.

### 2.3. Transcriptome Analysis Results

#### 2.3.1. Summary of Transcriptome Sequencing

To gain insight into the molecular mechanisms underlying the hypocotyl elongation of pakchoi, we conducted transcriptome sequencing using the hypocotyls of mutant *lhy7.1* and WT at the time point with the highest elongation rate (72 h after sowing) using three biological replicates, labelled as WT-1, WT-2, and WT-3 for WT hypocotyls and *lhy7.1-1*, *lhy7.1-2*, and *lhy7.1-3* for *lhy7.1* hypocotyls. As shown in Table 1, the RNA sequencing of six cDNA libraries generated 38.04 Gb of clean data. The Q30 base percentage of each sample was not less than 93.80%, while the GC content was between 46.99% and 47.73%. Mapping the clean reads of each sample to the designated reference genome showed that the mapping efficiency was greater than 87.76%. About 84.02% of the clean reads were mapped uniquely and used for subsequent analysis (Table 1). Gene expression levels were estimated with the fragments per kilobase of exon per million fragments mapped (FPKM) values. The Pearson’s correlation coefficient (Pearson’s r value) between each two samples was calculated based on the FPKM values; the correlation and cluster analysis results between any two samples were shown in the form of a heat map (Figure 3A). Additionally, R software was used for a principal component analysis (PCA) to evaluate sample repeatability and correlation (Figure 3B). From Figure 3, it can be seen that the three biological replicates of each treatment had strong repeatability and the differences between treatments were significant, indicating the reliability of the transcript data in this study. There were 13,249, 23,706, 36,373, 32,044, 36,580, and 49,354 obtained clean reads which were further annotated to main databases, including COG, KOG, Pfam, Swiss-Prot, eggNOG, and NR (Figure 3C). By comparison with WT, the DEGs in *lhy7.1* were identified with DEGseq2 software using a false discovery rate (FDR) < 0.01 and |log2fold change| ≥ 2. A total of 4568 DEGs were detected, of which 44.96% (2054 genes) were up-regulated and 55.04% (2514 genes) were down-regulated in *lhy7.1* compared to WT (Figure 3D). The top five up-regulated DEGs were *Brassica_rapa_newGene_2066*, *BraC09g041160*, *BraC09g042290*, *BraC05g034830*, and *BraC01g039810* with a 9.26, 8.54, 8.01, 7.99, and 7.93 log2foldchange, respectively. The top five down-regulated DEGs were *BraC06g017380*, *BraC03g050740*, *Brassica_rapa_newGene_4506*, *BraC09g040990*, and *BraC05g035030* with −9.33, −8.90, −8.54, −8.52, and −8.30 log2foldchange, respectively (Figure 3D). These results demonstrated that the experimental samples and results were considered reliable for further analysis.

#### 2.3.2. GO Enrichment Analysis of DEGs

GO analysis is commonly used to comprehensively describe the properties of genes and gene products. In our study, a total of 3665 DEGs were annotated to GO items, which were divided into three major functional categories: biological process (BP), cellular component (CC), and molecular function (MF). The results of GO classification indicated that the DEGs were significantly enriched in 20 terms in the BP, MF, and CC categories (Figure 4). In the BP category, significant GO terms were the “photosynthesis”, “protein-chromophore linkage”, “photosynthesis, light harvesting”, “carbon fixation”, “photosystem II assembly”, “lignin catabolic process”, and “carbohydrate metabolic process” (Figure 4A). A total of 41 DEGs were annotated to “response to auxin” (Figure 4A). In the MF category, the remarkably enriched GO terms were “chlorophyll binding”, “copper ion binding”, and “hydroquinone: oxygen oxidoreductase activity “(Figure 4B). In the CC category, the top seven enriched GO terms were “chloroplast thylakoid membrane”, “chloroplast”, “photosystem II”, “photosystem II oxygen evolving complex “, “photosystem I”, “chloroplast thylakoid”, and “chloroplast part” (Figure 4C).

#### 2.3.3. Kyoto Encyclopedia of Genes and Genomes (KEGG) Pathway Analysis

A KEGG pathway analysis was performed on the DEGs (Figure 5). We mapped the DEGs obtained between *lhy7.1* and WT into KEGG pathways, among which 1676 DEGs were mapped into 128 metabolic pathways. The top 20 KEGG pathways were shown in Appendix A. The main KEGG categories were: “plant-pathogen interaction” (ko04626, 9.07%), “plant hormone signal transduction” (ko04075, 8.53%), “carbon metabolism” (ko01200, 6.68%), “photosynthesis” (ko00195, 4.77%), “starch and sucrose metabolism” (ko00500, 4.77%), “phenylpropanoid biosynthesis” (ko00940, 4.53%), etc.

The DEGs were used for KEGG pathway enrichment analysis to corroborate the biological pathways activated in *lhy7.1* and WT. The top five enriched pathways among the up-regulated DEGs in *lhy7.1* were involved in “ABC transporters” (ko02010), “phenylpropanoid biosynthesis” (ko00940), “fatty acid elongation” (ko00062), “flavonoid biosynthesis” (ko00941), and “plant hormone signal transduction” (ko04075) (Figure 5A); whereas the top five enriched pathways among down-regulated DEGs were associated with “photosynthesis” (ko00195), “photosynthesis–antenna proteins” (ko00199), “carbon fixation in photosynthetic organisms”(ko00710), “glyoxylate and dicarboxylate metabolism”(ko00630), and “carbon metabolism” (ko01200) (Figure 5B). We focused on the pathways of “plant hormone signal transduction” (n = 84), “photosynthesis” (n = 80), “photosynthesis–antenna proteins” (n = 30), and “carbon fixation in photosynthetic organisms” (n = 44). These results demonstrated that “plant hormone signal transduction” and the photosynthetic system may play major regulatory roles in hypocotyl elongation in *lhy7.1*.

#### 2.3.4. Identification of Key Regulatory Genes Involved in Hypocotyl Elongation

To further explore the key regulatory genes in hypocotyl elongation, we analyzed DEGs that were enriched in “plant hormone signal transduction”, “photosynthesis–antenna proteins”, and the “photosynthesis” pathways. The network of the “plant hormone signal transduction” pathway revealed 84 up-regulated DEGs that were significantly enriched (Figure 5A). Interestingly, a total of 40 DEGs were identified in the auxin pathway, with *SAUR10*, *SAUR19-24*, *SAUR50*, *SAUR61*, *SAUR64*, *IAA3*, *IAA6*, *AUX1*, *GH3.1*, *GH3.12*, *GH3.17*, *X10A*, *ARG7*, *LAX2*, *LAX3*, *ARF8*, and *AFB3* markedly up-regulated in *lhy7.1* (Figure 6A). In addition, *GID1B* was also significantly up-regulated in the GA pathway (Figure 6B).

There were 30 and 80 down-regulated DEGs that were significantly enriched for “photosynthesis–antenna proteins” and “photosynthesis” pathways, respectively (Figure 6C,D). The gene expression levels of *LHCA1*, *LHCA2*, *LHCA3*, *LHCA4*, *LHCA5*, and *LHCA6* encoding light-harvesting complex I proteins, and *LHCB1.3*, *LHCB2.4*, *LHCB3*, *LHCB4.1*, *LHCB4.2*, *LHCB4.3*, *LHCB5*, and *LHCB7* encoding light-harvesting complex II proteins, were down-regulated, indicating a decrease in the light capture efficiency of *lhy7.1*. Additionally, the photosynthesis of *lhy7.1* was further inhibited (Figure 6C). The genes involved in photosynthetic system I (*PSAB*, *PSAD1*, *PSAE1*, *PSAF*, *PSAG*, *PSAH*, *PSAH1*, *PSAK*, *PSAL*, *PSAN*, and *PSAO*) and photosynthetic system II (*PSBA*, *PSB27-1*, *PSB28*, *PSBK*, *PSBP*, *PSBQ2*, *PSBR*, *PSBS*, *PSBY*, and *PSBW*), were down-regulated in *lhy7.1* (Figure 6D). In addition, all the genes in photosynthetic electron transport were down-regulated. The gene *PETC*, encoding the Cytb6/f complex, was down-regulated as well. Owing to the reduction in light capture efficiency, the photosynthesis processes in *lhy7.1* were also affected.

#### 2.3.5. qRT-PCR Validation

To verify the reliability and repeatability of the RNA-seq results, a quantitative real-time PCR (qRT-PCR) was performed to measure the expression of six DEGs that were randomly selected from the top 20 enriched KEGG pathways. *BraC01g000200* is involved in the “carbon fixation in photosynthetic organisms” pathway. *BraC01g040730* and *BraC06g009990* are involved in the “photosynthesis” pathway. *BraC01g032200* and *BraC05g045610* are involved in the “plant hormone signal transduction” pathway. *BraC02g002040* is involved in the “ABC transporters” pathway. *BraC02g002040*, *BraC01g032200*, and *BraC05g045610* were up-regulated and *BraC01g000200*, *BraC01g040730*, and *BraC06g009990* were down-regulated, consistent with the RNA-seq data (Figure 7, Appendix A). These results confirm the reliability of the RNA-seq results and reflect the actual transcriptome changes in *lhy7.1*.

### 2.4. Proteome Analysis Results

#### 2.4.1. Differentially Expressed Protein (DEP) Analysis and Subcellular Localization

To further study the molecular mechanisms of hypocotyl elongation in pakchoi, label-free proteomic techniques were used to analyze the differences in protein abundance between *lhy7.1* and WT. The results of PCA showed that the samples were repeatable and there was a significant difference between *lhy7.1* and WT (Appendix A). In this study, the proteins with *p* < 0.05 and FC ≥ 1.5 and FC ≤ 0.67 were defined as DEPs. Compared with WT, 1007 proteins were differentially abundant in *lhy7.1*, with 392 (38.93%) up-regulated and 615 (61.07%) down-regulated (Figure 8A,B). The top five up-regulated DEPs were BraC09g008890.1, BraC03g031110.1, BraC05g004120.1, BraC09g007480.1, and BraC09g061360.1 with 14.46, 13.87, 13.38, 13.30, and 10.96 log2foldchange, respectively. The top five down-regulated DEPs were BraC09g017310.1, BraC09g009030.1, BraC03g027410.1, BraC07g024090.1, and BraC09g040950.1 with −13.51, −13.42, −12.69, −12.05, and −11.26 log2foldchange, respectively (Figure 8B). Since there were many more down-regulated DEPs than up-regulated DEPs, it suggested that there was a tendency to inhibit relevant protein expression in *lhy7.1*.

Subcellular localization is one of the most important aspects in determining protein functions. We predicted the subcellular location of the DEPs using CELLO software. According to the predicted results, DEPs were mainly located in the cytoplasm (258, 25.62%) and chloroplasts (224, 22.24%; Figure 8C). The active DEPs in chloroplasts, the main site of photosynthesis in higher plants, might affect the hypocotyl elongation of *lhy7.1*, compared to WT.

#### 2.4.2. GO and KEGG Enrichment Functional Analyses of DEPs

The DEPs were further annotated to functional categories of GO terms and KEGG pathways. Using a GO analysis, the DEPs enrichment was noted in biological processes, cellular components, and molecular functions (Figure 9A). In BPs, there were mainly 47, 331, and 340 DEPs that were significantly enriched in “photosynthesis” (GO: 0015979), “metabolic process” (GO: 0008152), and “cellular process” (GO: 0009987), respectively. Among the CCs, there were 138, 138, and 178 DEPs that were significantly enriched in “chloroplast part” (GO: 0044434), “plastid part” (GO: 0044435), and “chloroplast” (GO: 0009507), respectively. In terms of MFs, 18, 71, and 262 DEPs were significantly enriched in “ribosome RNA binding” (GO: 0019843), “oxidoreductase activity” (GO: 0016491), and “catalytic activity” (GO: 0003824) (Figure 9A). The top three significantly enriched KEGG pathways were “metabolic pathways”, “biosynthesis of secondary metabolites”, and “biosynthesis of amino acids” (Figure 9B). We focused on the pathways of “photosynthesis”, “photosynthesis–antenna proteins”, and “carbon fixation in photosynthetic organisms” (Figure 9B). These results were similar to the KEGG enrichment analysis of the transcriptome analysis results.

### 2.5. Association Analysis of the Transcriptome and Proteome Results

A combined analysis of the transcriptome and proteome was performed to reveal the correlations between DEGs and DEPs. The genes detected in the proteome and transcriptome were divided into nine modules according to their expression patterns and presented in a nine quadrant diagram (Figure 10A). There were 9184 genes detected in both the proteome and the transcriptome in *lhy7.1* vs. WT. A total of 1896 DEGs or DEPs, including 691 up-and 1205 down-regulated, were obtained in quadrants three and seven (Figure 10A). The results showed that not all mRNA/protein ratios reflected the corresponding changes in transcription and protein levels due to the complex relationship between mRNA and protein expression levels.

To further explore the biological pathways which the DEGs and DEPs play roles in, a KEGG pathway enrichment analysis was conducted. Therefore, the DEGs and DEPs in quadrants three and seven indicating a positive correlation between protein abundance and transcript accumulation were analyzed. The KEGG enrichment analysis showed that photosynthesis-related pathways were highly enriched by DEGs and DEPs (Figure 10B,C), such as “photosynthesis”, “photosynthesis–antenna proteins”, “carbon fixation in photosynthetic organisms”, etc.

## 3. Discussion

In pakchoi, the hypocotyl is an important agronomic trait. Seedlings with a long hypocotyl were more suitable for mechanical harvesting during production. We found a spontaneous mutant of pakchoi, *lhy7.1*, which possesses a long hypocotyl. The average hypocotyl length of *lhy7.1* was significantly longer than that of WT (Figure 1D,E) and the hypocotyl elongation rate in *lhy7.1* was significantly higher than that of WT at any time point (Figure 1F). Similar results were observed in a long hypocotyl mutant of *Arabidopsis* [27]. Previous studies have also shown that the hypocotyl elongation rate in the long hypocotyl mutant *elh1* was significantly higher than that of the short hypocotyl WT at any time point in cucumber [28]. Studies of long-hypocotyl mutants, such as *lhy7.1*, may provide a theoretical basis for the breeding of long-hypocotyl varieties which could be applied in mechanized production.

Cell elongation, rather than cell division, largely contributes to hypocotyl growth [29,30]. Previous studies have shown that hypocotyl growth mainly depends on cell longitudinal elongation in *Arabidopsis* and cucumber [5,6,28,29]. We examined the length of the hypocotyl cells of both WT and *lhy7.1* grown in an artificial climate box under controlled environmental conditions using semi-thin sections. The longitudinal length of hypocotyl cells in *lhy7.1* was significantly longer than that of WT (Figure 1G–I). These results indicated that hypocotyl length was primarily determined by cell longitudinal elongation in pakchoi.

The elongation of hypocotyls is an important process in plant growth and development; the morphogenesis of hypocotyls is influenced by the synergistic effects of internal and external factors. Light is one of the most critical external environmental signaling factors for plants as it provides a source of energy for plant growth and development. In *Arabidopsis*, hypocotyl elongation was inhibited under red, far-red, blue, and UVB lights, compared with the dark conditions [7]. This study found that the hypocotyl elongation of WT and *lhy 7.1* was inhibited under white, red, and blue lights. The length of *lhy 7.1* was always longer than that of WT, except under dark conditions. However, red light promoted *lhy7.1* hypocotyl elongation compared to *lhy7.1* under white light (Figure 2A,B). Meanwhile, *lhy7.1* not only had significantly longer hypocotyls than WT, but also slowed down the elongation of the apical hook. In addition, most *lhy7.1* cotyledons did not fully unfold on the sixth DAS, while WT cotyledons had already rapidly unfolded after excavation, exhibiting a phenotype of partially dark morphogenesis (Figure 2C). The results indicated that *lhy7.1* exhibited a similar shading response under white light, likely due to the mutation.

A transcriptome analysis plays an important role in the study of pakchoi molecular mechanisms [31,32]. For example, an RNA-seq detected a key gene, *BcTT8*, for the regulation of anthocyanin synthesis [33]. This study found that DEGs were primarily enriched in “plant hormone signal transduction” (up-regulated genes), “photosynthesis”, “photosynthesis–antenna proteins”, and “carbon fixation in photosynthetic organisms” (down-regulated genes) (Figure 5). Further, we used label-free proteome sequencing technology to analyze the DEPs and found that they were mainly enriched in “photosynthesis”, “photosynthesis–antenna proteins”, and “carbon fixation in photosynthetic organisms” (Figure 9). These results were similar to the KEGG enrichment analysis of the DEGs. This study showed that the relationship between mRNA and protein expression levels is not completely identical (Figure 10A), which suggests that these proteins may have undergone post-transcriptional alteration and activation, releasing active proteins immediately without the requirement for the transcription of the relevant genes [34]. In this study, we focused on detecting the DEGs/DEPs of the same trend in the transcriptome and proteome which were enriched in photosynthesis-related pathways (Figure 10B,C), indicating that the mutant led to changes in the photosynthetic system. The analysis of DEG-enriched KEGG pathways showed that “plant hormone signal transduction” was the most remarkable and photosynthesis-related pathways were the most prominent KEGG pathways for DEGs and DEPs. Compared with the photosynthesis-related pathways, phytohormones were considered to play important functions in the molecular signaling for hypocotyl elongation in plants.

Auxin regulates growth and developmental processes by being engaged in cell division and growth, tissue differentiation, organ development, and a range of physiological responses throughout the whole lifetime of all plants [35]. Auxin also has regulatory functions by promoting the expression of early auxin-responsive genes such *AUX/IAA*, *Gretchen Hagen* (*GH3*), and *small growth hormone up-regulated RNA* (*SAUR*) genes [36]. In our study, 84 up-regulated DEGs were significantly enriched in the “plant hormone signal transduction” (ko04075) pathway and a total of 40 DEGs were identified in the auxin pathway (Figure 5A and Figure 6A). Interestingly, some genes associated with *AUXIN-RESISTANT 1* (*AUX1*), *GH3*, and *SAUR* were significantly up-regulated in *lhy7.1*. *AUX1*, an auxin influx carrier gene, belongs to a family of proton-driven transporter amino acid infiltration enzymes that transport auxin across the cell membrane. The hypocotyl elongation of *aux1* was reported to be inhibited compared to WT [37]. IAA3/SHY2 (SHORT HYPOCOTYL 2) mediates PIF-dependent hypocotyl elongation [38]. Many *GH3* family genes are involved in hypocotyl growth, like *DFL2* and *YDK1* [39,40]. Here, auxin associated genes *AUX1*, *IAAs*, *ARF8*, and the *GH3* family had increased expression in *lhy 7.1*, implying that changes in the auxin pathway occurring in *lhy 7.1* may be responsible for the longer hypocotyls due to the important roles of auxin in hypocotyl growth.

Gibberellins (GAs) could promote hypocotyl elongation mainly by degrading DELLA protein and require BZR1 and PIFs, which are key elements of GAs signal transduction. DELLA combines with the GA receptor GID1 and E3 ubiquitin ligase to form SCFSLY1/GID2 to induce polyubiquitination and the degradation of DELLA [41]. Here, *GID1B* was significantly up-regulated in *lhy7.1* (Figure 6B), suggesting that it may influence the synthesis of gibberellin and, thus, the changes in hypocotyl length in pakchoi.

Plants synthesize carbohydrates through photosynthesis, which starts from light capture. In our present study, the KEGG pathway enrichment analysis indicated that there were significant differences in the “photosynthesis–antenna proteins” and “photosynthesis” pathways between WT and *lhy7.1*. Among the DEGs, the genes encoding light-harvesting complex I and II, photosystem I and II, photosynthetic electron transport, and Cytb6/f complex proteins were down-regulated (Figure 6C,D). Additionally, a proteome analysis revealed the down-regulation of light-harvesting proteins CAB1, LHCB4.2, and LHCB4.3, as well as PsaF and PsaN in photosystem I, and PsbO2, PsbR, and PPL1 in photosystem II, PetA in the Cytb6/f +complex, and ATPC1 and ATPD in F-type ATPase.

Many previous studies have highlighted that photosynthesis and auxin-related pathways regulate hypocotyl elongation. For example, in *Arabidopsis*, hypocotyl length was promoted after the disruption of IAA3, which is a negative regulator of auxin signaling [38]. Das et al. (2016) observed that ethylene and shade treatments increased hypocotyl length compared to the control, with many auxin and photosynthesis-related genes having altered their expression levels. *SAUR10*, *SAUR19*, *SAUR50*, and *SAUR 64* were up-regulated [42]. Moreover, in cucumber, DEGs between a longer hypocotyl mutant and WT were annotated to “photosynthesis” and “phytohormone signal transduction” pathways [28]. These results supported our results that hypocotyl elongation may be due to the changes in photosynthesis and auxin pathways, indicating that the molecular mechanism of hypocotyl elongation may be conserved in higher plants. Our findings provide a certain reference value for further studies of hypocotyl elongation.

Based on the above experimental results, the photosynthetic system was further inhibited in *lhy7.1*, including stronger inhibition of light capture and photosystems I and II. Therefore, we speculated that these changes may be caused by the mutation, which deserves to be further studied. As auxin and light signaling are commonly integrated in hypocotyl development, we posit that *lhy 7.1* affects a complex network comprising correlated genes, ultimately promoting hypocotyl elongation.

## 4. Materials and Methods

### 4.1. Plant Materials and Growing Conditions

A spontaneous mutant of pakchoi possessing a long hypocotyl (*B*. *campestris* var. *communis* Tesn et Lee), *lhy7.1*, was identified in a wild-type (WT) background. WT is a green petiole-type inbred line with a short hypocotyl. Both WT and *lhy7.1* were planted in an experimental station of Shanghai Academy of Agricultural Sciences located in 2901 Beidi road, Minhang strict, Shanghai (121.33° E, 31.23° N).

The pakchoi inbred lines *lhy7.1* and WT were grown in an artificial climate box (RDN- 1000C, Yanghui, Ningbo, China) under controlled environmental conditions (14 h light/10 h dark photoperiod, 25 °C/20 °C and 70–75% humidity, 10,000 Lx light intensity) in the Shanghai Academy of Agricultural Sciences, China. The length and elongation rate of the hypocotyls of all plants were visually scored every 24 h after germination; data from at least 40 seedlings of each genotype were collected. Hypocotyl length was measured using vernier calipers.

To investigate the effect of light quality on hypocotyl elongation, the seedlings from *lhy7.1* and WT were cultured in an artificial climate box (14 h light/10 h dark photoperiod, 25 °C/20 °C and 70–75% humidity, 10,000 Lx light intensity) with red (R) and blue LED light sources with peak wavelengths of 620 and 420 nm, respectively. White light and dark treatment was also employed in parallel with these experiments. Hypocotyl length was measured at 7 d after sowing.

### 4.2. Microscopic Observation of the Hypocotyl Cells

To observe the histological changes in pakchoi hypocotyl development, semi-thin sections were prepared from WT and *lhy7.1* seedlings. Firstly, hypocotyls of 10-day-old seedlings were cut into 1 mm^3^ pieces and fixed at 4 °C for preservation and transportation. Tissues were post fixed with 1% OsO_4_ in 0.1 M PB (pH 7.4) for 2 h at room temperature in the dark. After removing OsO_4_, the tissues are rinsed in 0.1 M PB (pH 7.4) 3 times, 15 min each, followed by dehydration at room temperature through a series of different ethanol concentrations (30%, 50%, 70%, 80%, 95%, 100%, 100%, each for 20 min) and made transparent in acetone for 15 min. Then, resin penetration and embedding were as follows: 1:1 acetone/EMBed 812 for 2–4 h at 37 °C; 1:2 acetone/EMBed 812 overnight at 37 °C; pure EMBed 812 for 5–8 h at 37 °C. Then, the pure EMBed 812 was poured into the embedding models and the tissues were inserted and incubated in a 37 °C oven overnight. The embedding models with resin and samples were moved into a 65 °C oven to polymerize for more than 48 h. The resin blocks were then cut into 1.5 μm thin sections using a semi-thin slicer and the tissues were fished out onto the microscope slides. Toluidine blue dye solution was kept in an oven at 60 °C for 1h and then used to stain the slides for 2 min. The slides were washed with running water, differentiated with 95% alcohol, and the color was monitored under a light microscope. The slides were kept in an oven to dry and then covered with neutral resin. Lastly, well-stained sections were sealed with resin and coverslips for imaging with a microscope. Cell length was calculated from 30 cells of each hypocotyl using Image-Pro Plus 6.0 software (Media Cybemetics, Houston, TX, USA).

### 4.3. RNA Extraction, cDNA Library Construction and Sequencing

The hypocotyls of WT and *lhy7.1* seedlings 72 h after sowing were harvested for RNA sequencing with three biological replicates for each sample. The total RNA extraction, cDNA library construction, sequencing, and data analyses were conducted by Beijing Biomarker Technologies Co., Ltd. (Beijing, China) using an Illumina NovaSeq 6000 sequencing platform. Reads with adapters or poly-N or of low-quality were first removed using in-house Perl scripts to obtain clean reads. At the same time, Q20, Q30, GC-content, and sequence duplication level of the clean data were calculated. The fragments per kilobase of transcript per million fragments (FPKM) value of each gene was calculated using cufflinks [43] and read counts of each gene were obtained by htseq-count [44]. The differentially expressed genes (DEGs) with a false-discovery rate (FDR) < 0.01 and fold change (FC) ≥ 2 were identified using DESeq2 (Version 1.6.3) [45]. The FDR is calculated by subtracting the *p*-value from the error detection rate [46].

Functional enrichment analyses were then performed for these DEGs. Gene Ontology (GO) (http://geneontology.org/, accessed on 12 January 2022) enrichment was implemented by the GOseq R packages based on Wallenius non-central hyper-geometric distribution [47], which can adjust for gene length biases in DEGs. The Kyoto Encyclopedia of Genes and Genomes (KEGG) enrichment analysis of DEGs was conducted using the KOBAS software (2.0) [48]. Enrichment analysis was performed using the BMK Cloud (www.biocloud.net, accessed on 8 February 2022) online platform.

### 4.4. Proteome Analysis

Samples for protein extraction were collected as for transcriptome with three replicates. The sample extraction, sequencing, and data analysis were carried out by Biomarker Technologies (Beijing, China). The samples were ground into powder in liquid nitrogen and dissolved in 400 μL lysis buffer (4% SDS, 100 mM Tris-HCl, 100 mM dithiotreitol (DTT), at pH 7.6). The proteins collected at the filter were quantified using the bicinchoninic acid (BCA) protein assay kit (Beyotime Biotech Inc, Shanghai, China) according to manufacturer’s instruction. The absorbance was recorded at 562 nm using a Multiskcan FC spectrophotometer (Thermo Scientific, Waltham, MA, USA). For each sample, 2 μg of total peptides were separated and analyzed with a nano UPLC (EASY-nLC1200, Thermo Scientific, Waltham, MA, USA) coupled to a Q Exactive HFX Orbitrap instrument (Thermo Fisher Scientific, Waltham, MA, USA) with a nano-electrospray ion source.

The quantification of proteins was carried out using Proteome Discoverer software (Version 2.4.0.305, Thermo Fisher Scientific, Waltham, MA, USA). The protein sequence of *B*. campestris was retrieved through the reference genome of a non-heading Chinese cabbage (NHCC001 V1.0) [2]. The proteins with *p* < 0.05 and fold change (FC) ≥ 1.5 (up-regulated) and FC ≤ 0.67 (down-regulated) were defined as differentially expressed proteins (DEPs), which were further analyzed with GO and KEGG enrichment analysis. The protein subcellular localization prediction was performed using CELLO software (http://cello.life.nctu.edu.tw/) [49]. The visualized results as volcano plots and heatmaps and those from PCA, GO, and KEGG, were obtained using the R script.

### 4.5. Quantitative Real Time PCR

Total RNA from each sample was extracted using Transzol (Transgene, Beijing, China). The first strand cDNA was synthesized using the PrimeScript RTMaster Mix (Perfect Real Time) (TaKaRa, Shiga, Japan). The PCR reaction using TransStart Top Green qPCR SuperMix (+Dye I) was performed using a QuantStudio 5 Real-Time PCR Instrument (96-well 0.2 mL block) (Thermo Fisher Scientific, Waltham, MA, USA). *GAPDH* was used as a reference gene. Primer sequences are listed in Appendix A.

## 5. Conclusions

The mechanism of hypocotyl elongation was studied by phenotypic, cytological, transcriptome, and proteome analyses comparing *lhy7.1* and WT pakchoi (Figure 11). Cytological observations showed that the longer hypocotyl of *lhy7.1* may be due to cell longitudinal elongation. Furthermore, the DEGs and DEPs were determined to identify the key genes in response to auxin and were found to be significantly up-regulated; these genes included *AUX1*, *GH3.1*, *GH3.12*, *GH3.17*, *IAA3*, *IAA6*, *SAUR19–24*, etc. Conversely, the key genes and proteins of the photosynthetic system pathway were significantly down-regulated. Thus, the photosynthetic system was further inhibited in *lhy7.1*, compared to the WT. We concluded that the mutation in *lhy 7.1* caused expression changes of the genes involved in the auxin pathway and photosynthesis, finally leading to a longer hypocotyl. These findings will provide a new perspective for revealing the molecular mechanisms of hypocotyl elongation and a theoretical basis for the breeding of mechanized harvesting varieties of pakchoi.

## Figures and Tables

**Figure 1 ijms-24-13808-f001:**
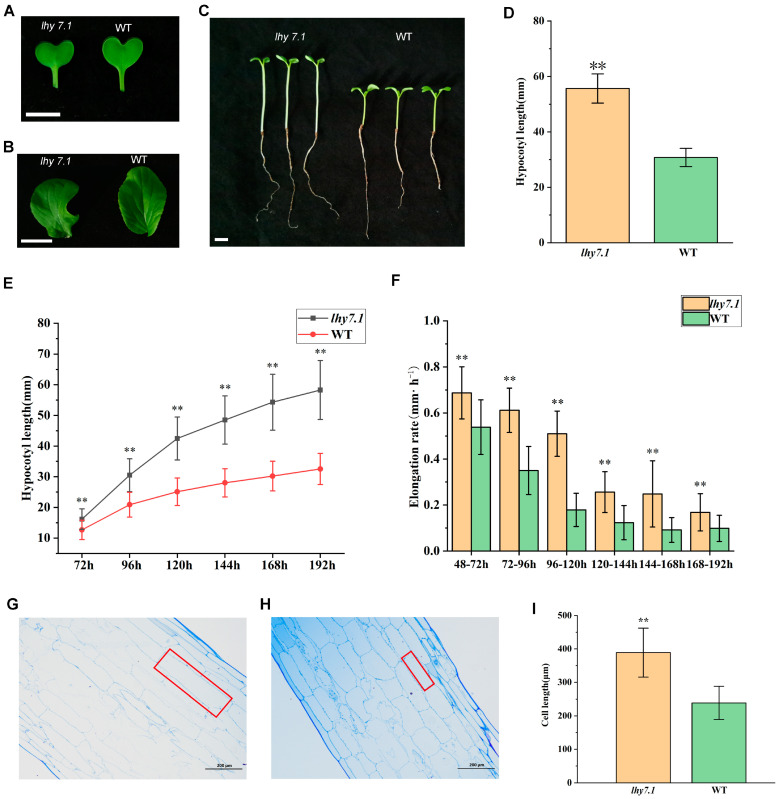
Phenotypic characterization of elongated hypocotyl mutant *lhy7.1* and wild-type (WT) pakchoi. (**A**,**B**) leaf colors of the mutant *lhy7.1* (left) and WT (right) at cotyledon and first true leaf stages. Scale bar = 1 cm. (**C**) hypocotyl phenotypes of *lhy7.1* (left) and WT (right), respectively. Scale bar = 1 cm. (**D**) hypocotyl lengths in *lhy7.1* and WT seedlings at 7 day after sowing (DAS). 40 seedlings per genotype (n = 40). Data shown are mean ± standard error (SE). Double asterisks indicate *p* < 0.01 (two-tailed Student’s *t*-test). (**E**,**F**) time-course analysis of hypocotyl elongation in *lhy7.1* and WT seedlings in pakchoi. (**E**) mean hypocotyl length and (**F**) mean hypocotyl elongation rate of the seedlings (n = 40 seedlings, respectively). Data shown are mean ± SE. Double asterisks indicate *p* < 0.01 (two-tailed Student’s *t*-test). (**G**) the longitudinal section of hypocotyl cells in *lhy7.1* at 10 DAS (×100), scale bar = 200 µm. The red box indicates cell size. (**H**) the longitudinal section of hypocotyl cells in WT at 10 DAS (×100), scale bar = 200 µm. The red box indicates cell size. (**I**) cell lengths of hypocotyls in *lhy7.1* and WT seedlings at 10 DAS. 30 cells per plant (n = 30). Data shown are mean ± SE. Double asterisks indicate *p* < 0.01 (two-tailed Student’s *t*-test).

**Figure 2 ijms-24-13808-f002:**
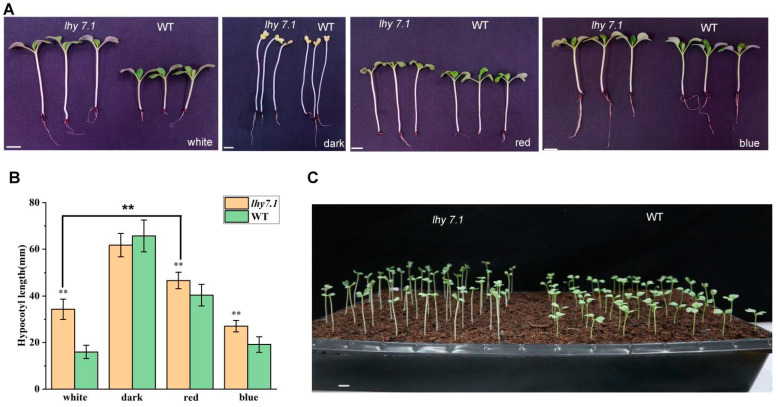
Hypocotyl growth of *lhy7.1* and WT in different light quality conditions. (**A**) the hypocotyl phenotype of *lhy7.1* (**left**) and WT (**right**) under white, dark, red, and blue lights at 7 DAS. (**B**) hypocotyl length in *lhy7.1* and WT under white, dark, red, and blue lights. The error bars represent standard error of the means. Double asterisks indicate *p* < 0.01 (two-tailed Student’s *t*-test). (**C**) the hypocotyl phenotype of *lhy7.1* (**left**) and WT (**right**) at 6 DAS under white light. Scale bar = 1 cm.

**Figure 3 ijms-24-13808-f003:**
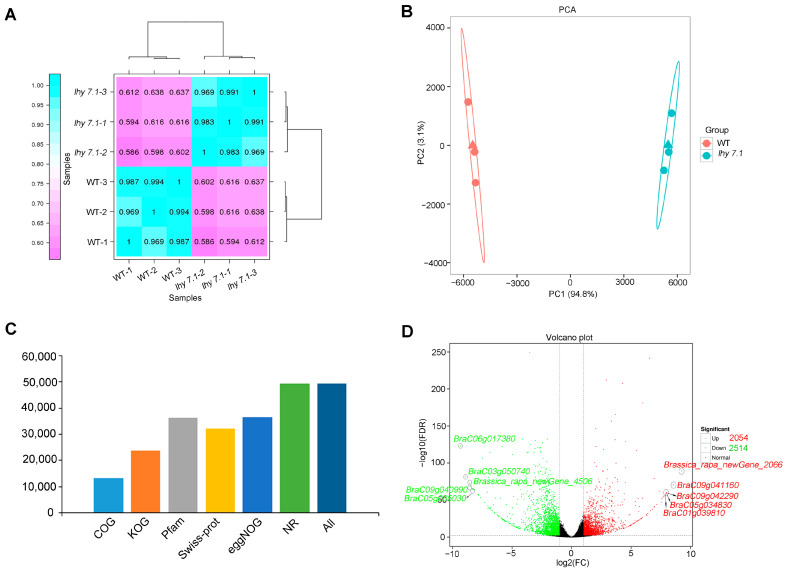
Summary of RNA-seq results. (**A**) heatmap of Pearson’s correlation coefficient (Pearson’s r value). Purple and blue boxes indicate Pearson’s r value between two samples. The closer r^2^ is to 1, the greater the correlation. (**B**) principal component analysis (PCA) of all samples to assess data quality. (**C**) numbers of clean reads annotated to Clusters of Orthologous Groups (COG), EuKaryotic Orthologous Groups (KOG), Pfam, SwissProt, Evolutionary Genealogy of Genes: Non-supervised Orthologous Groups (eggNOG), and Non-Redundant Protein Sequence (NR) databases. (**D**) volcano plots of the number of differentially expressed genes (DEGs) identified by the transcriptome analysis in *lhy7.1* compared with WT. Each point represents a gene. The *x*-axis represents the log2(foldchange) of DEGs and *y*-axis represents −log10(FDR). The green and red dots represent down-regulated and up-regulated genes, respectively. *Brassica_rapa_newGene_2066*, *BraC09g041160*, *BraC09g042290*, *BraC05g034830*, and *BraC01g039810* are the top five up-regulated genes; *BraC06g017380*, *BraC03g050740*, *Brassica_rapa_newGene_4506*, *BraC09g040990*, and *BraC05g035030* are the top five down-regulated genes.

**Figure 4 ijms-24-13808-f004:**
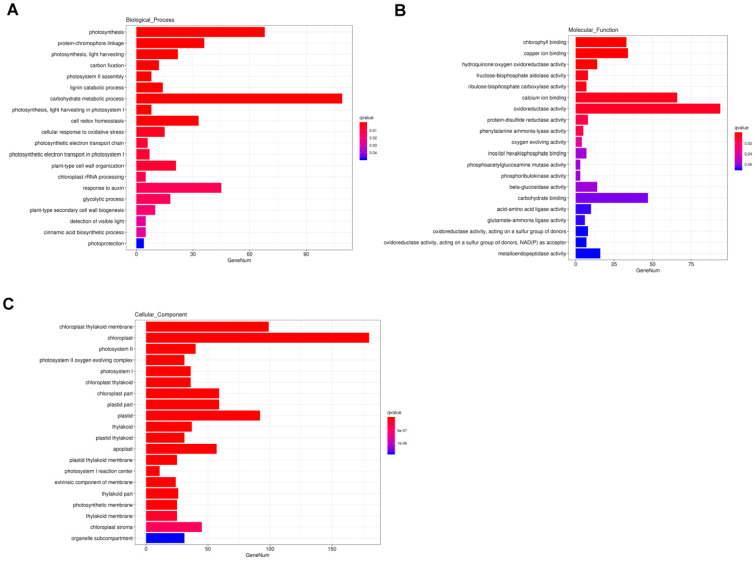
Enriched Gene Ontology (GO) terms among DEGs. (**A**) biological process (BP); (**B**) molecular function (MF); (**C**) cellular component (CC). Note: The *x*-axis indicates the number of DEGs involved in each pathway and the *y*-axis represents the name of the pathway. Bar color represents q value.

**Figure 5 ijms-24-13808-f005:**
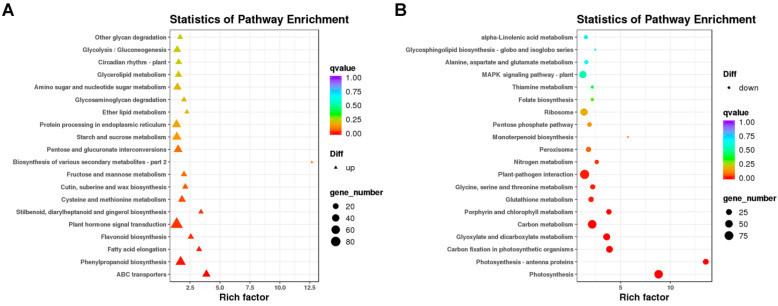
Kyoto Encyclopedia of Genes and Genomes (KEGG) enrichment analysis of DEGs. (**A**) enriched KEGG pathways of up-regulated DEGs. (**B**) enriched KEGG pathways of down-regulated DEGs. Note: The *x*-axis represents rich factors and the *y*-axis represents pathway names. The point and triangle sizes represent the number of DEGs involved, with larger points and triangles indicating more genes. Point and triangle colors indicates q value; the more significance, the redder the color.

**Figure 6 ijms-24-13808-f006:**
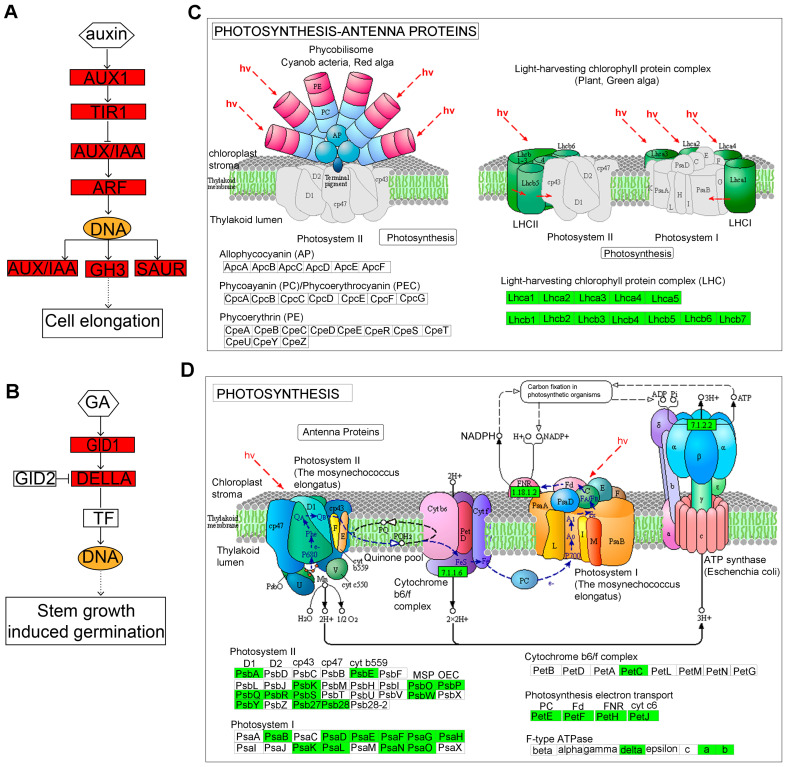
The DEGs involved in “plant hormone signal transduction” (**A**,**B**), “photosynthesis–antenna proteins” (**C**), and “photosynthesis” (**D**) pathways. (**A**) auxin signaling; (**B**) GA signaling pathways; (**C**) “photosynthesis–antenna proteins”; (**D**) “photosynthesis” pathways. Red indicates significantly up-regulated DEGs. Green indicates significantly down-regulated DEGs.

**Figure 7 ijms-24-13808-f007:**
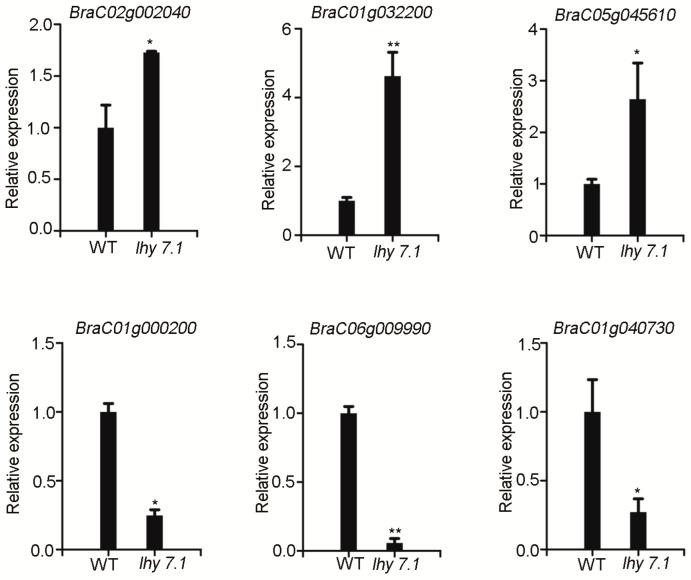
qRT-PCR analysis. Data shown represent average ± SE of three biological replicates. * *p* < 0.05, ** *p* < 0.01 (two-tailed Student’s *t*-test). *GAPDH* was used as reference gene for normalization.

**Figure 8 ijms-24-13808-f008:**
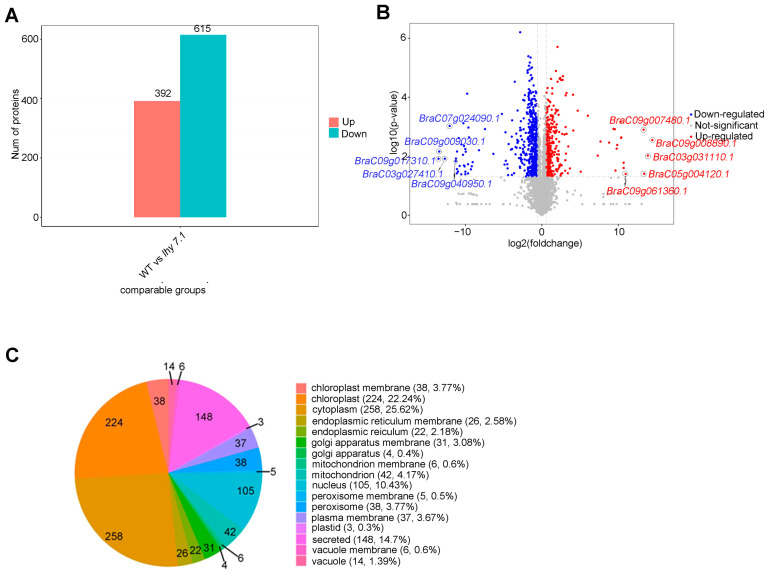
Analysis of differentially expressed proteins (DEPs) of *lhy7.1* and WT. (**A**) the distribution of DEPs. (**B**) volcano plot of DEPs. Blue dots represent down-regulated proteins; red dots represent up-regulated proteins; grey dots represent proteins that are not significantly differentially expressed. BraC09g008890.1, BraC03g031110.1, BraC05g004120.1, BraC09g007480.1, and BraC09g061360.1 are the top five up-regulated DEPs and BraC09g017310.1, BraC09g009030.1, BraC03g027410.1, BraC07g024090.1, and BraC09g040950.1 are the top five down-regulated DEPs. (**C**) subcellular localization analysis of DEPs.

**Figure 9 ijms-24-13808-f009:**
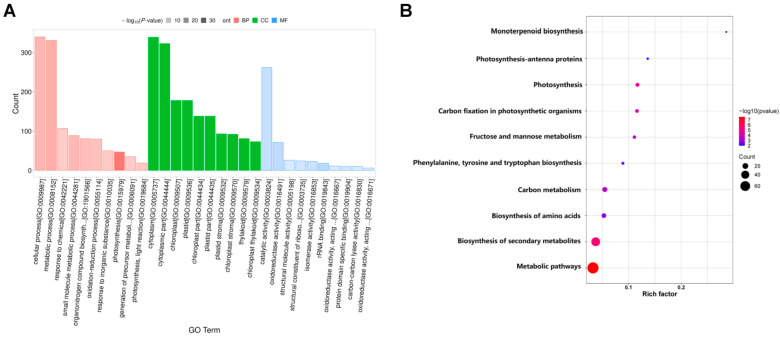
The functional annotation of DEPs. (**A**) GO classification and enrichment of DEPs. *X*-axis: GO term names; red bars represent the GO terms of biological processes; green bars represent the GO terms of cellular components; blue bars represent the GO terms of molecular functions. *Y*-axis: the number of DEPs annotated to related term. (**B**) KEGG pathways of DEPs. Each point represents a KEGG pathway. *Y*-axis: pathway names; *X*-axis: rich factor. Point colors represent significance; the redder the color, the greater the significance.

**Figure 10 ijms-24-13808-f010:**
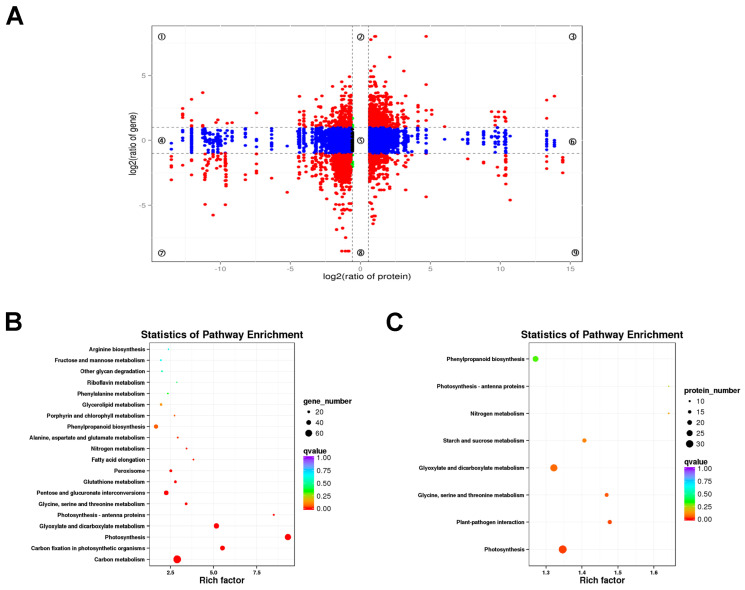
Combined analysis of DEGs and DEPs. (**A**) a nine quadrant diagram of protein and mRNA associations. Each point represents one gene. Quadrants 1 and 9 indicate the genes negatively correlated with proteins. Quadrants 3 and 7 show the genes positively correlated with proteins. Quadrants 1, 2, and 4 indicate that the protein abundance was lower than the RNA abundance. Quadrant 5 shows that the proteins and RNAs were similarly expressed without differences between WT and *lhy7.1*. Quadrants 6, 8, and 9 indicate the genes in which the abundance was lower than protein abundance. (**B**) KEGG enrichment analysis of DEGs with same expression trend as DEPs. (**C**) KEGG enrichment analysis of DEPs with the same expression trend as DEGs. The rich factor represents the ratio of DEGs annotated to the pathway to all genes annotated to the pathway.

**Figure 11 ijms-24-13808-f011:**
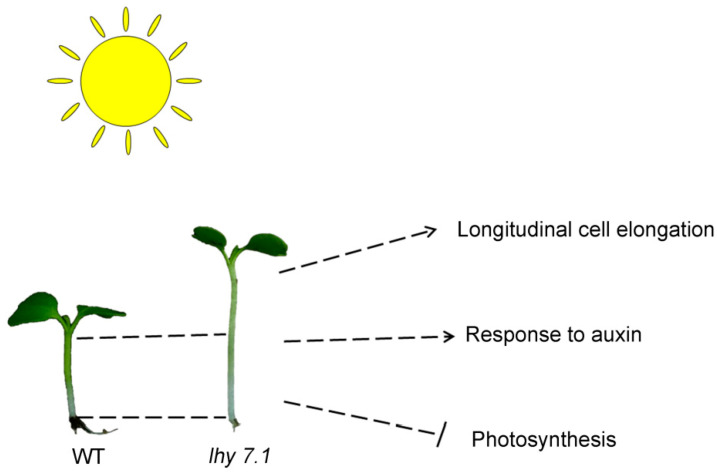
The proposed mechanism of hypocotyl elongation in *lhy7.1* of pakchoi based on cytology, transcriptome, and proteome data. The arrows represent the positive correlations of longitudinal cell elongation and “response to auxin” pathway with hypocotyl elongation while photosynthesis negatively correlated with hypocotyl elongation.

**Table 1 ijms-24-13808-t001:** Statistic of RNA-seq data for each sample.

Samples	Total Clean Reads	Clean Reads	Mapped Reads	Unique Mapped Reads (%)	GC Content (%)	Q30 (%)
WT-1	44,040,902	22,020,451	39,643,566 (90.02%)	38,125,106 (86.57%)	46.99	94.68
WT-2	38,548,448	19,274,224	33,831,669 (87.76%)	32,386,509 (84.02%)	47.53	93.81
WT-3	47,041,512	23,520,756	41,826,754 (88.91%)	40,171,957 (85.40%)	47.24	94.16
*lhy7.1*-1	40,919,018	20,459,509	37,127,273 (90.73%)	35,753,814 (87.38%)	47.65	93.80
*lhy7.1*-2	41,214,806	20,607,403	36,540,515 (88.66%)	35,216,896 (85.45%)	47.63	94.04
*lhy7.1*-3	42,684,144	21,342,072	38,598,224 (90.43%)	37,184,460 (87.12%)	47.73	93.89

Total clean reads: the number of clean reads, calculated on a single-ended basis. Clean reads: total number of pair-end reads in the clean data. Mapped reads: the number of reads mapped to the reference genome and the percentage of reads in the clean reads. Unique Mapped Reads: counts of reads mapped to a unique position in reference genome and proportion of that in clean data. GC content: GC content in clean data; that is, the percentage of bases G and C in the total bases in the clean data. Q30%: The percentage of bases whose clean data mass value was greater than or equal to 30.

## Data Availability

The transcriptomic sequencing data presented in this study are openly available in NCBI under accession no. SRR 36865760-36865765. The associated BioProject is PRJNA1002875.

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
