# Peer review of "Integrated Transcriptome and Proteome Analysis Revealed the Regulatory Mechanism of Hypocotyl Elongation in Pakchoi"

_ijms, 2023, doi:10.3390/ijms241813808_

Round 1
Reviewer 1 Report
Comments
1) Table 1. What do you mean by total clean reads and clean reads? How about raw reads?
2) Figure 3 legends are not well explained. Include details of analysis including statistics. Avoid short forms.
3) Figure 3D. Volcano plot, name at least 5-10 most significant up and down regulated genes. It will make figure more meaningful.
4) Figure 8B. In volcano plot, name at least 5-10 most significant up and down regulated genes. It will make figure more meaningful.
5) Figure 4. Keep top 5 most significant terms only.
6) Did you obtain permission from Kanehisa Laboratories for using KEGG figure?
7) Authors need to re-write figure legends. Avoid short forms.
8) Figure 11. What these arrows denotes. Again, not much information in figure legend.
9) Line 584. What parameters for DEGs selection?
10) The paper needs extensive revision for English language.
11) Line 19-20. Need rephrasing.
The paper needs extensive revision for English language.
Reviewer 2 Report
In this manuscript, the authors investigate the molecular mechanisms underlying stem elongation in B. campestris var. communis Tesn et Lee. This spontaneous mutant of pakchoi has a mutation in the lhy7.1 gene. The authors examined histologically/anatomically the maturation and growth of stem cells and the signaling involved. In particular, the authors focused on the expression of auxin and gibberellin regulated genes in the two WT and mutant groups. Finally, the authors also evaluated protein expression in the two groups. All analyses were performed in a controlled environment and with different light stimuli. The interesting results showed how light regulates the elongation process by modulating the hormonal and thus the genetic and protein response.
The manuscript is very clear and linear. The statistics used are correct and the data are well presented.
I would suggest that the authors improve the quality of Figures 6c and 6d.
I would also suggest that the authors add more ecological details about the mutant cultivar: where does it grow? Is it widespread?
Finally, can the data collected on pakchoi be extended to all plant species? The transferability to other plant species should be better discussed.
Round 2
Reviewer 1 Report
Authors has sufficiently addressed comments raised by. I endorse paper for publication.
Its fine now.